

# Direct aortic route versus transaxillary route for transcatheter aortic valve replacement: a systematic review and meta-analysis

Hsiu-An Lee[1],*, I-Li Su[1],*, Shao-Wei Chen[1], Victor Chien-Chia Wu[2], Dong-Yi Chen[2], Pao-Hsien Chu[2], An-Hsun Chou[3], Yu-Ting Cheng[1], Pyng-Jing Lin[1] and Feng-Chun Tsai[1]

[1] Department of Thoracic and Cardiovascular Surgery, Chang Gung Memorial Hospital, Linkou Medical Center, Chang Gung University, Taoyuan, Taiwan
[2] Department of Cardiology, Chang Gung Memorial Hospital, Linkou Medical Center, Chang Gung University, Taoyuan, Taiwan
[3] Department of Anesthesiology, Chang Gung Memorial Hospital, Linkou Medical Center, Chang Gung University, Taoyuan, Taiwan
* These authors contributed equally to this work.

Corresponding author
Feng-Chun Tsai,
m8293@cgmh.org.tw

## ABSTRACT

**Background:** The transfemoral route is contraindicated in nearly 10% of transcatheter aortic valve replacement (TAVR) candidates because of unsuitable iliofemoral vessels. Transaxillary (TAx) and direct aortic (DAo) routes are the principal nonfemoral TAVR routes; however, few studies have compared their outcomes.

**Methods:** We performed a systematic review and meta-analysis to compare the rates of mortality, stroke, and other adverse events of TAx and DAo TAVR. The study was prospectively registered with PROSPERO (registration number: CRD42017069788). We searched Medline, PubMed, Embase, and Cochrane databases for studies reporting the outcomes of DAo or TAx TAVR in at least 10 patients. Studies that did not use the Valve Academic Research Consortium definitions were excluded. We included studies that did not directly compare the two approaches and then pooled rates of events from the included studies for comparison.

**Results:** In total, 31 studies were included in the quantitative meta-analysis, with 2,883 and 2,172 patients in the DAo and TAx TAVR groups, respectively. Compared with TAx TAVR, DAo TAVR had a lower Society of Thoracic Surgery (STS) score, shorter fluoroscopic time, and less contrast volume use. The 30-day mortality rates were significantly higher in the DAo TAVR group (9.6%, 95% confidence interval (CI) = [8.4–10.9]) than in the TAx TAVR group (5.7%, 95% CI = [4.8–6.8]; $P$ for heterogeneity <0.001). DAo TAVR was associated with a significantly lower risk of stroke in the overall study population (2.6% vs. 5.8%, $P$ for heterogeneity <0.001) and in the subgroup of studies with a mean STS score of ≥8 (1.6% vs. 6.2%, $P$ for heterogeneity = 0.005). DAo TAVR was also associated with lower risks of permanent pacemaker implantation (12.3% vs. 20.1%, $P$ for heterogeneity = 0.009) and valve malposition (2.0% vs. 10.2%, $P$ for heterogeneity = 0.023) than was TAx TAVR.

**Conclusions:** DAo TAVR increased 30-day mortality rate compared with TAx TAVR; by contrast, TAx TAVR increased postoperative stroke, permanent pacemaker implantation, and valve malposition risks compared with DAo TAVR.

## INTRODUCTION

Transcatheter aortic valve replacement (TAVR) enables the safe and effective treatment of inoperable or high-surgical-risk patients with severe aortic valve disease, without using a cardiopulmonary bypass (*Kodali et al., 2012*; *Makkar et al., 2012*). Randomized controlled trials have demonstrated that TAVR is an effective alternative to surgical aortic valve replacement in intermediate-risk patients (*Leon et al., 2016*; *Reardon et al., 2017*).

Transcatheter aortic valve replacement is more favorable than surgical aortic valve replacement when using transfemoral (TF) access (*Gargiulo et al., 2016*), which is thus used as the default approach for performing TAVR in numerous institutions. However, peripheral vascular occlusion, stenosis, calcification, or tortuosity precludes TF access in approximately 10% of patients (*Grover et al., 2017*), necessitating the use of an alternative route, such as transapical (TA), transaxillary (TAx), direct aortic (DAo), and transcarotid routes. To select the optimal treatment technique in patients unsuitable for TF TAVR, clinicians need to understand the outcomes of using different nonfemoral routes.

Transapical route was the first alternative TAVR route developed for patients with unsuitable iliofemoral vessels (*Grover et al., 2017*; *Walther et al., 2015*). However, the procedure is associated with relatively high rates of bleeding, ventricular damage (*Al-Attar et al., 2009*), myocardial injury (*Ribeiro et al., 2015*), and mortality (*Fröhlich et al., 2015*; *Panchal et al., 2014*). DAo and TAx routes are also principal alternatives to TF; both have results comparable to those of the TF route (*Adamo et al., 2015*; *Arai et al., 2016*; *Chandrasekhar et al., 2015*; *Fröhlich et al., 2015*). However, data comparing the outcomes of using the DAo and TAx TAVR routes are limited. Therefore, we conducted this systematic review and meta-analysis to compare the morbidity and mortality associated with these two approaches.

## MATERIALS AND METHODS

### Literature search

This systematic review of published studies was performed following the Preferred Reporting Items for Systematic Reviews and Meta-Analyses (PRISMA) guidelines, with a PRISMA checklist provided as Table S1. This study has been prospectively registered with PROSPERO (registration number: CRD42017069788). A computerized search of the Medline, PubMed, Embase, and Cochrane databases was performed to identify all relevant

studies published before December 31, 2019 by using the following keywords: "transcatheter," "aortic valve," "TAVR," "TAVI," "direct aortic," "transaortic," "transaxillary," "axillary," "trans-subclavian," and "subclavian." The exact string of keywords is reported in Supplemental Material 1. Review articles or meta-analyses were not included for analysis, but their citations and references were searched for additional relevant studies. Citations were screened at the title and abstract levels and retrieved as a full report if outcome data of TAVRs were provided. Two evaluators (H.A. Lee and S.W. Chen) independently searched for and reviewed the articles. Discrepancies were discussed and resolved through consensus.

## Study selection

Inclusion criteria were as follows: (1) original article in English with full-length content available, (2) at least 10 consecutive patients who underwent either DAo or TAx TAVR, (3) outcomes defined using the Valve Academic Research Consortium (VARC) definition (as VARC-1 or VARC-2) (Kappetein et al., 2012; Leon et al., 2011), and (4) separate results for patients undergoing DAo TAVR or TAx TAVR. Exclusion criteria were as follows: (1) overlapping patients or subgroup studies of the main study, (2) studies that focused on the valve-in-valve procedure, (3) studies that focused on TAVR combined with another procedure, and (4) the use of devices other than Medtronic CoreValve (MCV; Medtronic, Minneapolis, MN, USA) and Edwards Valve (EV; Edwards Lifesciences, Irvine, CA, USA). Studies that did not directly compare the 2 approaches were also included. The most recent publications were retained when two or more similar studies were reported by the same institution or author.

## Data extraction

Relevant information was collected by H.A. Lee and S.W. Chen. The study-level characteristics extracted were first author name, publication year, study type (e.g., single-centered or multicentered), number of studies, location, study period, route (DAo or TAx), patient number, and VARC version (Table 1). The arm-level characteristics items extracted included age, logistic EuroSCORE, Society of Thoracic Surgery (STS) score, comorbidities, previous cardiac surgery, left ventricular ejection fraction, and devices (Table 2). Data on the primary and secondary outcomes for either DAo or TAx were also collected.

## Outcome measures

The primary outcomes were 30-day stroke and mortality rates after TAx or DAo TAVR. These results were further stratified by mean STS scores of <8 and ≥8 after TAx or DAo TAVR. The 30-day stroke rates after MCV and EV use were also compared. If a study did not report the 30-day mortality or stroke rates, in-hospital mortality or stroke rates were used. The secondary outcomes were device success, conversion to traditional surgery, valve malposition, acute kidney injury, major bleeding, major vascular complication, new permanent pacemaker (PPM) implantation, paravalvular leakage (PVL) grade of ≥2, 30-day cardiovascular mortality, and 1-year mortality.

**Table 1 Study data.**

| First author | Year | Locations/ country | Study type | No. of centers | Study period | Access | Patient number | VARC |
|---|---|---|---|---|---|---|---|---|
| Khan | 2018 | US | Single center | 1 | 2013–2015 | TAx, DAo | 51 | 2 |
| Damluji | 2018 | US, France | Multi-center | 3 | 2008–2017 | TAx, DAo | 84 | 2 |
| Fiorina | 2016 | Italy | Multi-center | 4 | 2007–2014 | TAx, DAo | 147 | 2 |
| Adamo | 2015 | Italy | Single center | 1 | 2007–2014 | TAx, DAo | 32 | 2 |
| Zhan | 2020 | US | Single center | 1 | 2015–2018 | TAx | 10 | 2 |
| Dahle | 2019 | US | Multi-center | NA | 2015–2018 | TAx | 1249 | 2 |
| Hysi | 2019 | France | Single center | 1 | 2015–2017 | TAx | 43 | 2 |
| Gleason | 2018 | US | Multi-center | 45 | NA | TAx | 202 | 1 |
| Terzian | 2017 | France | Single center | 1 | 2006–2014 | TAx | 36 | 1 |
| Schäfer | 2017 | Germany | Multi-center | 2 | 2010–2016 | TAx | 100 | 2 |
| Laflamme | 2014 | Canada | Single center | 1 | 2010–2012 | TAx | 18 | 2 |
| Muensterer | 2013 | Germany | Single center | 1 | 2007–2011 | TAx | 40 | 2 |
| Testa | 2012 | Italy | Single center | 1 | NA | TAx | 70 | 1 |
| Gilard | 2012 | France | Multi-center | 34 | 2010–2011 | TAx | 184 | 1 |
| Romano | 2019 | France | Single center | 1 | 2011–2014 | DAo | 265 | 2 |
| Cocchieria | 2019 | Eurpoe | Multi-center | 18 | 2013–2015 | DAo | 253 | 2 |
| D'Ancona | 2019 | German | Single center | 1 | 2012–2014 | DAo | 106 | 2 |
| Petzina | 2017 | Germany | Single center | 1 | 2012–2014 | DAo | 99 | 2 |
| Bruschi | 2017 | Europe | Multi-center | 9 | 2012–2014 | DAo | 92 | 2 |
| Bonaros | 2017 | Europe | Multi-center | 18 | 2013–2015 | DAo | 301 | 2 |
| Ropponen | 2016 | Finland | Single center | 1 | 2008–2014 | DAo | 36 | 1 |
| Arai | 2016 | France | Single center | 1 | 2011–2014 | DAo | 289 | 2 |
| Wendt | 2015 | Germany | Single center | 1 | 2012–2014 | DAo | 30 | 1 |
| Thourani | 2015 | US | Multi-center | NA | 2011–2014 | DAo | 868 | 2 |
| Ribeiro | 2015 | Canada | Single center | 1 | 2007–2015 | DAo | 45 | 2 |
| Ramlawi | 2015 | US | Single center | 1 | 2011–2015 | DAo | 78 | 2 |
| Okuyama | 2015 | US | Single center | 1 | 2007–2014 | DAo | 51 | 2 |
| Jagielak | 2015 | Poland | Multi-center | NA | 2013–2014 | DAo | 32 | 2 |
| Bruschi | 2015 | Italy | Single center | 1 | 2008–2013 | DAo | 50 | 2 |
| Spargias | 2014 | Greece | Single center | 1 | NA | DAo | 25 | 1 |
| Dahle | 2014 | Norway | Single center | 1 | 2009–2013 | DAo | 30 | 1 |

**Note:**
Basic information of studies included in the meta-analysis. DAo, direct aortic; NA, not available; VARC, Valve Academic Research Consortium; TAx, transaxillary; US, the United States.

## Quality assessment

We assessed the quality of the included studies by using the Newcastle-Ottawa Scale (NOS) (*Wells et al., 2014*). Quality scores ranged from 0 (lowest) to 8 (highest). The NOS was

**Table 2 Baseline and procedural characteristics of patients.**

| Variable | DAo patients | | TAx patients | |
|---|---|---|---|---|
| | Available data, $n$ | Weight mean ± SD | Available data, $n$ | Weight mean ± SD |
| Age, year | 2,236 | 82.7 ± 1.2 | 2,136 | 80.0 ± 1.7 |
| Logistic EuroSCORE | 642 | 22.0 ± 7.1 | 681 | 22.6 ± 5.3 |
| STS score | 1,161 | 7.5 ± 1.8 | 1,737 | 8.9 ± 3.0 |
| Old Stroke, % | 1,957 | 12.1 | 1,957 | 12.1 |
| Atrial fibrillation, % | 1,152 | 28.6 | 590 | 39.0 |
| Peripheral arterial disease, % | 2,035 | 41.6 | 1,919 | 64.4 |
| Chronic kidney disease, % | 1,946 | 28.5 | 557 | 16.0 |
| Previous cardiac surgery, % | 1,961 | 23.6 | 870 | 23.5 |
| Left ventricular ejection fraction, % | 1,965 | 62.3 ± 9.7 | 311 | 52.1 ± 1.6 |
| Device (CoreValve, %) | 2,847 | 23.2 | 1,852 | 29.3 |

**Note:**
DAo, direct aortic; TAx, transaxillary; SD, standard deviation.

applied to each article separately by H.A. Lee and S.W. Chen and disagreements were resolved by consensus between the 2 reviewers.

## Statistical analysis

This meta-analysis included studies that did not directly compare the two approaches and pooled rates of events from the included studies for comparison. Random-effects models were used to pool the estimates of primary and secondary outcomes from individual studies for each arm (TAx or DAo). In contrast to a fixed-effects model, a random-effects model enables the true underlying effect to vary among individual studies. $I^2$ above 25%, 50%, and 75% were considered to represent low, moderate, and high heterogeneity across the studies, respectively (*Higgins et al., 2003*). The pooled estimates between TAx and DAo TAVR were compared using mixed-effects models. Statistical significance was set at $P < 0.05$ with a two-tailed test. Data were analyzed using Comprehensive Meta-Analysis (version 2.2; Biostat, Englewood, NJ, USA).

## RESULTS

### Literature search

Our initial web-based literature search yielded 703 records. We screened the titles and abstracts of all 703 studies, of which 583 did not satisfy our inclusion and exclusion criteria. We downloaded and assessed the full-text of 120 articles for eligibility. After a review of the full-text articles, we excluded 19 articles that employed duplicated cohorts, 49 that did not report the outcomes of patients who received TAx or DAo TAVR, 13 that did not use VARC definitions for reporting the outcomes, one that included only TAVR with combined coronary artery bypass grafting procedure, four that used devices other than Edward or Medtronic, one that enrolled <10 cases, and one that employed a valve-in-valve procedure (Supplemental Material 2, Table S2). Thus these 31 studies were used in the quantitative

meta-analysis (Fig. 1), with 2,883 and 2,172 patients in the DAo TAVR and TAx TAVR groups, respectively (*Adamo et al., 2015*; *Arai et al., 2016*; *Bonaros et al., 2017*; *Bruschi et al., 2017*, *2015*; *Cocchieri et al., 2019*; *D'Ancona et al., 2019*; *Dahle & Rein, 2014*; *Dahle, Kaneko & McCabe, 2019*; *Damluji et al., 2018*; *Fiorina et al., 2017*; *Gilard et al., 2012*; *Gleason et al., 2018*; *Hysi et al., 2019*; *Jagielak et al., 2015*; *Khan et al., 2018*; *Laflamme et al., 2014*; *Muensterer et al., 2013*; *Okuyama et al., 2015*; *Petzina et al., 2017*; *Ramlawi et al., 2015*; *Ribeiro et al., 2015*; *Romano et al., 2019*; *Ropponen et al., 2016*; *Schäfer et al., 2017*; *Spargias et al., 2014*; *Terzian et al., 2017*; *Testa et al., 2012*; *Thourani et al., 2015*; *Wendt et al., 2015*; *Zhan et al., 2020*).

## Quality assessment

The quality of the 31 studies included in the meta-analysis was assessed using NOS, scored in the range of 0–8 points. The NOS scores for all 31 studies ranged between 5 and 7 points, with a median score of 6 points (Table S3).

## Baseline and procedural characteristics

Table 2 presents the available baseline and procedural characteristics. The mean age of the DAo TAVR and TAx TAVR groups was 82.7 ± 1.2 and 80.0 ± 1.7 years, respectively. No substantial differences in logistic EuroSCORE (22.0 ± 7.1 in DAo vs. 22.6 ± 5.3 in TAx) and STS score (7.5 ± 1.8 in DAo vs. 8.9 ± 3.0 in TAx) were noted. The percentage of MCV use in the TAx TAVR group seemed to be higher than that in the DAo TAVR group.

## TAx and DAo TAVR outcomes

We analyzed the two primary outcomes, the 30-day stroke and mortality rates (Fig. 2). The 30-day mortality rates of the DAo TAVR and TAx TAVR groups were significantly different with mortality rates of 9.6% (95% confidence interval (CI) [8.4–10.9]) and 5.7% (95% CI [4.8–6.8]), respectively ($P$ for heterogeneity < 0.001). The pooled 30-day stroke rate in the DAo TAVR group (2.6%, 95% CI [1.9–3.4]) was significantly lower than that in the TAx TAVR group (5.8%, 95% CI [4.9–7.0]; $P$ for heterogeneity < 0.001).

We then analyzed secondary outcomes. Patients were more likely to require new PPM implantation after TAx TAVR (20.1%, 95% CI [15.5–25.6]) than after DAo TAVR (12.3%, 95% CI [9.4–16.0]; $P$ for heterogeneity = 0.009). Valve malposition occurred more frequently in patients who underwent TAx TAVR (10.2%, 95% CI [3.4–27.1]) than in patients who underwent DAo TAVR (2.0%, 95% CI [0.9–4.7]; $P$ for heterogeneity = 0.023). The conversion rate was higher in the DAo TAVR group (2.8%, 95% CI [2.1–3.6]) than in the TAx TAVR group (0.9%, 95% CI [0.6–1.6]; $P$ for heterogeneity < 0.001). No significant differences in the other secondary outcomes were identified between the two groups (Fig. 2).

# DISCUSSION

## TAx vs. DAo TAVR

TAx TAVR is the most commonly used percutaneous, nonfemoral approach that does not require general anesthesia or endotracheal intubation. TAx TAVR is also less invasive than

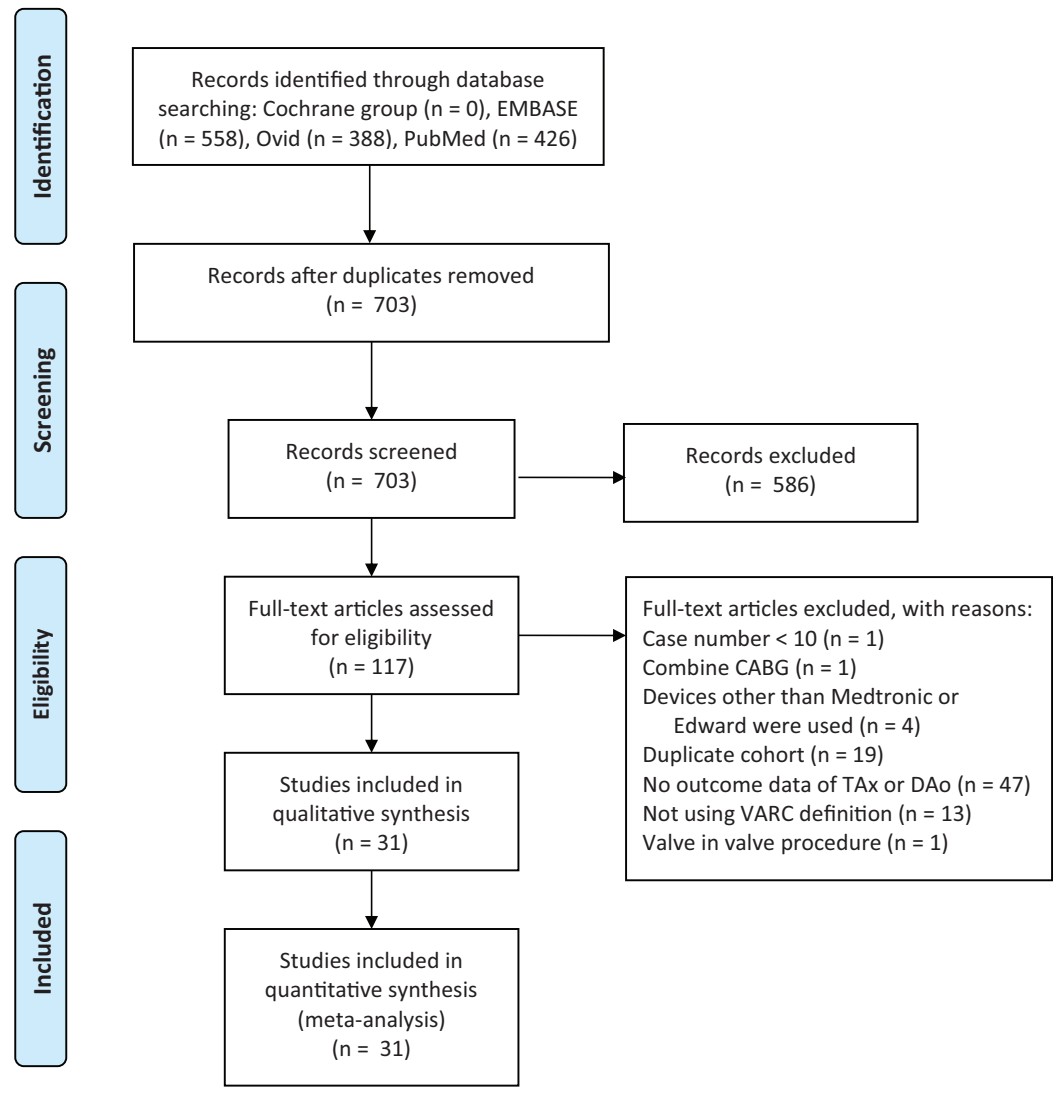

**Figure 1 Flow of study selection process.** CABG, coronary arterial bypass grafting; DAo, direct aortic approach; TAx, transaxillary; VARC, Valve Academic Research Consortium.

DAo and TA TAVR because it does not require entering the chest cavity, thereby reducing lung complication risks, thus shortening the ventilator time and intensive care unit stay.

In DAo TAVR, the delivery system enters directly through the ascending aorta, which requires minimal manipulation of the peripheral vessels, thereby reducing the incidence of peripheral vascular complications. Furthermore, cardiac surgeons are more familiar with DAo TAVR than with TA TAVR; therefore, DAo TAVR use may improve bleeding control and prevent myocardial injury, which can result in impaired ventricular function and ventricular pseudoaneurysm.

Data from more than 5,000 patients were analyzed in our systematic review and meta-analysis, which is the largest sample that has been used to compare TAx and DAo TAVR outcomes. We found that TAx TAVR was associated with a lower 30-day mortality rate,

| Outcome / group | No. of study | No. of event / Total number | | Event rate, % (95% CI) | $I^2$ (%) | P for heterogeneity |
|---|---|---|---|---|---|---|
| 30-Day mortality | | | | | | <0.001 |
| DAo | 19 | 209 / 2335 | | 9.6 (8.4–10.9) | 11.1 | |
| TAx | 15 | 119 / 2172 | | 5.7 (4.8–6.8) | 0.0 | |
| 30-Day CV death | | | | | | 0.933 |
| DAo | 8 | 32 / 715 | | 5.5 (3.6–8.3) | 51.7 | |
| TAx | 8 | 37 / 688 | | 5.6 (3.6–8.5) | 0.0 | |
| 1-year mortality | | | | | | 0.600 |
| DAo | 8 | 391 / 1493 | | 22.8 (18.2–28.1) | 70.7 | |
| TAx | 7 | 107 / 517 | | 20.7 (15.4–27.3) | 33.4 | |
| Stroke | | | | | | <0.001 |
| DAo | 15 | 44 / 2035 | | 2.6 (1.9–3.4) | 0.0 | |
| TAx | 11 | 104 / 1879 | | 5.8 (4.9–7.0) | 0.0 | |
| AKI | | | | | | 0.461 |
| DAo | 14 | 491 / 2177 | | 11.7 (6.8–19.3) | 94.2 | |
| TAx | 9 | 62 / 520 | | 8.2 (3.7–17.3) | 61.8 | |
| Major bleeding | | | | | | 0.283 |
| DAo | 10 | 83 / 580 | | 14.5 (7.5–26.3) | 86.5 | |
| TAx | 9 | 104 / 770 | | 8.5 (4.0–17.4) | 85.6 | |
| Major vascular complication | | | | | | 0.670 |
| DAo | 12 | 45 / 1147 | | 4.4 (2.7–6.9) | 0.0 | |
| TAx | 13 | 70 / 2096 | | 3.8 (2.3–6.2) | 70.9 | |
| PPM | | | | | | 0.009 |
| DAo | 15 | 161 / 1349 | | 12.3 (9.4–16.0) | 57.5 | |
| TAx | 13 | 338 / 2119 | | 20.1 (15.5–25.6) | 79.5 | |
| PVL grade 2/3 | | | | | | 0.113 |
| DAo | 13 | 76 / 1223 | | 7.3 (4.6–11.3) | 77.9 | |
| TAx | 11 | 86 / 834 | | 12.3 (7.7–19.2) | 59.2 | |
| Device unsuccess | | | | | | 0.943 |
| DAo | 15 | 73 / 1199 | | 6.6 (4.1–10.5) | 40.1 | |
| TAx | 11 | 91 / 1851 | | 6.8 (3.8–11.8) | 85.0 | |
| Conversion | | | | | | <0.001 |
| DAo | 10 | 50 / 1885 | | 2.8 (2.1–3.6) | 0.0 | |
| TAx | 7 | 12 / 1740 | | 0.9 (0.6–1.6) | 6.9 | |
| Valve malposition | | | | | | 0.023 |
| DAo | 7 | 5 / 488 | | 2.0 (0.9–4.7) | 0.0 | |
| TAx | 2 | 6 / 83 | | 10.2 (3.4–27.1) | 70.3 | |

Event rate, % (95% CI) — 0 10 20 30

**Figure 2 Forrest plot of TAx and DAo TAVR outcomes.** The pooled incidence of mortality, stroke, and other complications of TAx and DAo TAVR. DAo, direct aortic; TAVR, transcatheter aortic valve replacement; TAx, transaxillary.

compatible with the findings of previous studies (*Damluji et al., 2018*; *Fröhlich et al., 2015*). Moreover, TAx TAVR was associated with higher postoperative stroke and PPM implantation rates than was DAo TAVR. Studies have reported similar trends; however, statistical significance was not demonstrated in these studies, which may be due to insufficient sample sizes (*Adamo et al., 2015*; *Damluji et al., 2018*; *Fiorina et al., 2017*; *Fröhlich et al., 2015*).

## Stroke

Post-TAVR stroke occurrence remains a major concern and cause of increased morbidity and mortality. In the present meta-analysis, the stroke rate was higher in the TAx group than in the DAo group. The mechanism for the lower stroke rate after DAo TAVR is unclear. Transcranial Doppler studies have reported that cerebral embolism predominantly occurred during manipulation of the calcified aortic valve while prostheses were being positioned and implanted (*Kahlert et al., 2012*). The shorter distance and straight course between the device entry site (on the ascending aorta) and the aortic

annulus of the DAo route may enable surgeons to implant the stented valve more accurately and rapidly with less aortic valve manipulation, resulting in fewer distal embolisms.

In contrast to DAo TAVR, TAx TAVR involves the advancement of the delivery catheter from the right or left subclavian artery to the ascending aorta, thus traversing the origins of the vertebral artery, carotid artery, aortic arch, and ascending aorta, which may induce atherosclerotic plaques and cerebral embolism. The flow of the vertebral artery or right carotid artery may be compromised during the procedure, particularly when the diameter of the innominate artery or left subclavian artery is only marginally wider than the delivery catheter. Moreover, TAx TAVR can cause vessel wall disruption along the innominate and subclavian arteries and the origins of the carotid and vertebral arteries, resulting in enhanced thrombogenicity, which may be linked to stroke (*Barthélémy, Collet & Montalescot, 2016*).

### PPM implantation

A study comparing TAx and DAo TAVR in four high-volume Italian centers concluded that the TAx route is an independent predictor for increased PPM implantation (*Fiorina et al., 2017*)—compatible with our finding that PPM implantation rate was higher after TAx TAVR than after DAo TAVR. Implantation depth is known to be a predictor of PPM implantation after TAVR. DAo TAVR may provide better control over device placement than does TAVR with peripheral access, potentially resulting in better coaxial alignment and more accurate implant depth, thereby causing fewer conduction disturbances (*Bruschi et al., 2017*). Large-scale studies reported that TA TAVR was associated with a significantly lower rate of PVL than was TF TAVR (*Kodali et al., 2014*; *Van Belle et al., 2014*). This finding suggests that more direct routes, such as DAo or TA, provide increased device placement control and thus lower PVL and PPM implantation rates. Furthermore, our meta-analysis indicated that TAx TAVR was associated with significantly higher valve malposition and numerically higher PVL compared with DAo TAVR ($P = 0.113$).

### Study limitations

First, all studies included in our analysis were observational, and thus, heterogeneity between the 2 groups was inevitable. However, the 2 groups cannot be accurately balanced without undertaking a randomized controlled trial. Second, to ensure the standardization of the definitions of stroke and other complications, we enrolled only the studies that used the VARC definitions; however, this limited the number of patients analyzed, thereby reducing the power of the meta-analysis. Third, the pace of reporting does not match the rapid advancement of TAVR technology; therefore, these results may not represent the real outcomes of the most recent devices.

## CONCLUSIONS

The present meta-analysis compared the outcomes of DAo and TAx TAVR. We determined that the 30-day mortality rate was higher in patients who underwent DAo TAVR, but the postoperative stroke and PPM implantation rates were higher in patients who underwent

TAx TAVR. Our findings could help TAVR candidates with unsuitable femoral access optimize their selection of alternative access.

## ACKNOWLEDGEMENTS

The authors thank Alfred Hsing-Fen Lin for his assistance with the statistical analysis.

### Funding

The authors received no funding for this work.

### Competing Interests

The authors declare that they have no competing interests.

### Author Contributions

- Hsiu-An Lee performed the experiments, authored or reviewed drafts of the paper, and approved the final draft.
- I-Li Su performed the experiments, prepared figures and/or tables, and approved the final draft.
- Shao-Wei Chen conceived and designed the experiments, authored or reviewed drafts of the paper, and approved the final draft.
- Victor Chien-Chia Wu analyzed the data, authored or reviewed drafts of the paper, and approved the final draft.
- Dong-Yi Chen analyzed the data, prepared figures and/or tables, and approved the final draft.
- Pao-Hsien Chu conceived and designed the experiments, authored or reviewed drafts of the paper, and approved the final draft.
- An-Hsun Chou conceived and designed the experiments, authored or reviewed drafts of the paper, and approved the final draft.
- Yu-Ting Cheng analyzed the data, prepared figures and/or tables, and approved the final draft.
- Pyng-Jing Lin conceived and designed the experiments, authored or reviewed drafts of the paper, and approved the final draft.
- Feng-Chun Tsai conceived and designed the experiments, authored or reviewed drafts of the paper, and approved the final draft.

### Data Availability

The raw data is available in the Supplemental Files.

### Supplemental Information

Supplemental information for this article can be found online at http://dx.doi.org/10.7717/peerj.9102#supplemental-information.

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
