# Peer review of "Direct aortic route versus transaxillary route for transcatheter aortic valve replacement: a systematic review and meta-analysis"

_PeerJ, doi:10.7717/peerj.9102_

## Round 0.1 · original submission · Major Revisions

The reviewers' comments are critical to ensure the quality of the manuscript. Please make changes in a point-by-point style and do the same in the rebuttal letter.

Reviewer 1 ·

Basic reporting

Please see below

Experimental design

Please see below

Validity of the findings

Please see below

Additional comments

In the manuscript by Lee et al, entitled “Direct aortic versus transaxillary transcatheter aortic valve
replacement: a systematic review and meta-analysis”, the authors searched for studies reporting clinical outcomes after TAVI performed by direct aortic or transaxillary and analyzed 25 studies. Overall, mortality was similar, but post-procedural complications were higher with the transaxillary approach.

The registration on PROSPERO, the inclusion of studies with VARC definition and the exclusion of very small studies are important elements of this study.
Of course the study has important limitations, mainly the observational design of studies and the low number of patients nd events included.

The search was performed until July 2017. To make this study more interesting and appealing for the contemporary practice, please update it to Oct 2019. Few more studies will be added but will provide more power and more contemporary data.

Medtronic Corevalve and Edwards valves are the most used and for which we have the vast majority of strong evidence, but the authors should clarify the reason why they excluded studies with new valves. How many of such studies were excluded? It would be useful for readers to report a supplementary table with all excluded studies with the reasons and the reference.

Please report in the text or supplement the exact string of keywords used to do the search in each database.

Please add in the abstract the total number of patients analyzed in the 25 studies included.

Abstract conclusion: “both” is not necessary.

Please clarify in all the text including the abstract that the meta-analysis included studies not directly comparing the 2 approaches and reported and pooled rates of events from these studies.

Introduction: the authors state that transfemoral is the preferred approach but in some cases it is not possible. This is true, but the authors should clearly state for readers the reason for this preference. Indeed, transfemoral access is the one that allowed TAVI to be superior to SAVR (Gargiulo G et al, Ann Int Med 2016). In this meta-analysis, TAVI was non-inferior to SAVR and for the first time even in intermediate risk patients, but most of all, when TAVI was transfemorl it was superior in terms of mortality to SAVR. These data were then confirmed by subsequent analyses and trials. Please add this important point and citation.

Figure quality should be implemented.

A mother tongue revision of English language would be needed.

Reviewer 2 ·

Basic reporting

The references used are dated and mostly of only historical significance. The current manuscript is only about high-risk an inoperable patients.

Experimental design

no comment

Validity of the findings

no comment

Additional comments

The manuscript is well written and the methodology sound. The meta-analysis is dated and fails to reflect current practice in the US where TAVR is approved for all levels of risk. Discussing STS PROM <8 vs >8 is useless today.

·

Basic reporting

See below

Experimental design

See below

Validity of the findings

See below

Additional comments

Specific comments follow:

Summary: The authors started off with over 400 studies devoted to TAx TAVR and DAo TAVR. They used certain rules to select studies for a working poll. The final group consisted of 25 studies. From these studies, the authors looked at TAx TAVR and DAo TAVR outcomes. After considering different factors, the authors concluded that DAo TAVR patients had a better outcome than TAx TAVR patients.

Points:
1. The selected pool of studies is too small and is too selective. The selection scheme culled out over 90% of the original studies. The authors eliminated some records, because the study writers did not use precise definitions. The authors should have attempted to “translate” those studies into common terms. This would have enlarged the pool of useful studies. Thus the analysis would have been more useful.

2. The authors need to review their writing. The following are major issues:
a. In APA style, the punctuations follow citations. The authors had the order reversed.
b. In APA style, when there are three researchers on a paper, all three are listed. The authors did not follow this guidance for the one research paper that had three researchers.
c. In APA style, when the names of the researchers are made part of the sentence, then only the year is placed inside the citation parentheses. In one spot, the authors did not follow the APA guidance.
d. A voice should be selected and maintained. The authors switched back and forth from the first-person pronouns to the third-person pronouns.
e. Formal writing avoids anthropomorphizing usage. The authors used personal pronouns to refer to objects.
f. Parallel structure and a consistent pattern will help the reader to follow the discussion. The author did not do this with TAx TAVR and DAo TAVR. And the authors did not do this for sentences that contain a series.
g. Shorten forms should never be used before the long form. In several places the authors used the shorten form before using the long form.
h. Use the active voice. The authors had active voice, but in a few spots the authors used the passive voice.
i. Use strong verbs. The authors used weak verbs in a few places.
j. Avoid using long sentences with several subordinate clauses. The authors used very long sentences with several subordinate clauses.

---

## Round 0.2 · accepted · Accept

One of the reviewers still has a few minor points. I trust that you can make an easy fix during the proof production stage as very minor changes.

Reviewer 1 ·

Basic reporting

ok

Experimental design

ok

Validity of the findings

ok

Additional comments

The authors did a good job in addressing all comments raised by both reviewers. The manuscript is updated and has improved significantly.

·

Basic reporting

New Review

I am using the authors’ feedback to provide my comments.

See below

Experimental design

See below

Validity of the findings

See below.

Additional comments

I see that the authors revisited this area. Instead of beginning with 400-plus studies, they started off with 703 studies. After culling, they came up with about 32 useful studies. (I ran the math twice and came up with 32 both times. Double check the list of culled out records, because it appears that one is missing.)
== When I saw the word “finally” in line 149, I thought the authors were adding another 31 studies. It was not until I was in the “Quality Assessment” section that I realized that the sentence in line 149 was referring to the final group. I recommend using a different lead-in phrase such as:

Thus these 31 studies were used in the quantitative meta-analysis…..

I was impressed that these 31 studies looked at 5,055 patient outcomes. So the final pool of studies was a decent representation.

== In line 80, you stated that you looked at “all relevant studies published before July 2017.” In Table 1, there are eight studies that were published after 2017. You might wish to revise that sentence in line 80.

== From a quick read of the submission, I was able to follow the flow of their presentation. I was not tripped up by the writing. This is a vast improvement.